# Recent Advances of Taxol-Loaded Biocompatible Nanocarriers Embedded in Natural Polymer-Based Hydrogels

**DOI:** 10.3390/gels7020033

**Published:** 2021-03-24

**Authors:** Silvia Voci, Agnese Gagliardi, Roberto Molinaro, Massimo Fresta, Donato Cosco

**Affiliations:** 1Department of Health Sciences, University “Magna Græcia” of Catanzaro, Campus Universitario “S.Venuta”, I-88100 Catanzaro, Italy; silvia.voci@studenti.unicz.it (S.V.); gagliardi@unicz.it (A.G.); fresta@unicz.it (M.F.); 2IRCCS Ospedale San Raffaele srl, I-20132 Milan, Italy; molinaro.roberto@hsr.it

**Keywords:** cancer, hydrogels, liposomes, nanoparticles, polysaccharides, proteins, paclitaxel

## Abstract

The discovery of paclitaxel (PTX) has been a milestone in anti-cancer therapy and has promoted the development and marketing of various formulations that have revolutionized the therapeutic approach towards several malignancies. Despite its peculiar anti-cancer activity, the physico-chemical properties of PTX compromise the administration of the compound in polar media. Because of this, since the development of the first Food and Drug Administration (FDA)-approved formulation (Taxol^®^), consistent efforts have been made to obtain suitable delivery systems able to preserve/increase PTX efficacy and to overcome the side effects correlated to the presence of some excipients. The exploitation of natural polymers as potential materials for drug delivery purposes has favored the modulation of the bioavailability and the pharmacokinetic profiles of the drug, and in this regard, several formulations have been developed that allow the controlled release of the active compound. In this mini-review, the recent advances concerning the design and applications of natural polymer-based hydrogels containing PTX-loaded biocompatible nanocarriers are discussed. The technological features of these formulations as well as the therapeutic outcome achieved following their administration will be described, demonstrating their potential role as innovative systems to be used in anti-tumor therapy.

## 1. Introduction

### 1.1. Paclitaxel in Anti-Cancer Therapy

Many drugs currently being used in clinical practice are obtained from natural sources. Probably the most famous examples are the antibiotic penicillin and the aspirin precursor salicylic acid [1]. Paclitaxel (PTX) is one of the most successful antineoplastic agents coming from a natural source that is available today. It belongs to the Taxanes family and has a peculiar mechanism of action that has been employed to treat several types of malignancies [2,3,4]. Initially, it was employed for the treatment of refractory ovarian cancer; successively, it was approved for the cure of breast cancer and other tumors, alone or in association with other antineoplastic compounds, such as gemcitabine, bevacizumab, doxorubicin and cisplatin [5,6,7,8].

PTX was first isolated from the bark of trees of the *Taxus* genus (*Taxus brevifolia* and *Taxus baccata*), but its use was initially hampered because of both the non-ecological sustainability of the isolation procedure and the inadequate yield of the resulting compound [9]. To overcome these difficulties, in 1994, a semisynthetic approach involving acylation of the precursor 10-deacetylbaccatin III was provided [10,11,12], and since then, several other alternative methods for its development have been proposed, including microbial fermentation engineering [13,14]. PTX is chemically made up of an asymmetric diterpenoid 17-C skeleton composed of a four-membered taxane ring, followed by an oxetane ring and a side chain esterified at the C-13 position [15,16] (Figure 1a).

PTX is commonly known as the first microtubule-stabilizing agent able to arrest the cell cycle in the G_2_/M phase, promoting apoptotic cell death [17]. The compound binds the assembled microtubules in correspondence to the taxol-binding domain of β-tubulin, precluding their depolymerization [18] (Figure 1b). Structure–activity relationship studies provided the pharmacophoric model of PTX which consists of the taxane ring, the C-13 side chain and the benzoyl residue at C-2 [16,18]. Its bulky chemical structure, low aqueous solubility (<0.01 mg/mL) and, consequently, high lipophilicity (logP ~4) make its delivery a challenge [19]. Indeed, the absence of ionizable functional groups represents a serious drawback in this respect because modulation of the pH may promote the solubility of the drug in polar media, as has been true for other compounds such as doxorubicin [20,21,22] and vancomycin [23]. Regarding this, the active loading of a weak acid-modified PTX derivative, PTX-succinic acid (PTX-SA), was recently proposed [24]. Liposomes containing PTX-SA were characterized by a high entrapment efficiency (97% with respect to the initial 5 mg/mL added during the preparation procedure), a significant in vivo anti-cancer efficacy and negligible side effects when compared to commercial Taxol^®^ [24].

Since the discovery of PTX, several derivatives have been proposed in order to improve the physico-chemical features of the drug and circumvent the emerging resistance to taxols which has been induced by the β-tubulin isoforms (especially the isoform β-tubulin III) and by the expression of ATP-dependent efflux pumps, i.e., P-glycoprotein (P-gp). For example, the addition of two methoxy-groups at C7 and C10 exerted a poor affinity for P-gp, as was also true in the case of cabazitaxel (Jevtana^®^), the latest PTX derivative approved by the U.S. Food and Drug Administration (FDA). Moreover, the efficacy of the derivative Larotaxel against urethral bladder and breast cancers is under clinical investigation [25,26].

Considering the pharmacological relevance that PTX has had—and still has—in anti-cancer therapy, the aim of this mini-review is to highlight the main advances that have come about from the development of various carriers containing the active compound embedded in natural polymer-based hydrogels. The aim here is to describe the physico-chemical features of these formulations, discussing the applications of injectable in situ forming hydrogels for the delivery of PTX (Figure 2).

### 1.2. Overview on Marketed PTX-Loaded Nanosystems

The design/choice of an appropriate vehicle is an essential step in the administration of an active compound because it should not influence the pharmacological activity of the drug it carries, nor should it provoke side effects, thus avoiding off-target toxicity phenomena [27]. Some of the physico-chemical properties (i.e., lipophilicity) of anti-cancer drugs may raise a technological issue because they are administered by intravenous infusion, so a certain polar characteristic of the medium is required [28,29]. In this regard, a 1:1 *v*/*v* mixture made up of absolute ethanol and polyoxyethylated castor oil (Cremophor EL) was proposed as a suitable system for the parental administration of PTX, becoming the first marketed product of the drug (Taxol^®^, 6 mg/mL of PTX) [30]. However, severe side effects such as hypersensitivity reactions due to the release of 2-ethylhexyl-phthalate from the infusion tube, neutropenia and neuropathy resulted as a consequence of the organic solvents used, so premedication of patients with corticosteroids and antihistamines became mandatory [31]. However, it remains that these associations may induce modification of the PTX pharmacokinetic and pharmacodynamic profiles [32]. Moreover, the administration of Taxol^®^ was related to instability phenomena such as drug precipitation following dilution as well as the need for specific equipment [33].

The nanoencapsulation of a compound in a drug delivery system is a technological approach able to modulate the physico-chemical and pharmacological features of the molecule it transports [34,35,36]. The use of colloidal systems can promote increased drug solubility, stability, bioavailability and therapeutic efficacy while a decrease in the dose-related side effects can be expected because the administration times can be reduced [37,38,39,40,41].

In 2005, the FDA approved a nanoformulation developed aimed at overcoming the drawbacks related to the use of organic solvents. This nanoformulation was made up of PTX bound to endogenous serum protein albumin (Abraxane^®^ or nab-paclitaxel). It was approved as a first-line treatment for metastatic breast and pancreatic cancers as well as advanced non-small cell lung cancer [42]. The high-pressure homogenization procedure promoted reversible non-covalent albumin-PTX bonds and the formation of nanoparticles of 130 nm [43]. Besides the absence of toxic excipients, the main advantage of Abraxane^®^ was its increased accumulation in solid tumors due to the significant uptake of the formulation. This came about through the secreted protein acidic rich in cysteine (SPARC) and caveolin-mediated transcytosis pathway once the interaction between the albumin nanoparticles and glycoprotein gp60 had occurred [44,45]. Successively, two clinical trials based on the association of Abraxane^®^-carboplatin and Abraxane^®^-gemcitabine were approved by the FDA for the treatment of non-small cell lung cancer and metastatic pancreatic adenocarcinoma (NCT00540514 and NCT00844649) [46]. Abraxane^®^ increased the PTX therapeutic index and decreased the occurrence of Cremophor-related sensitivity; chemotherapy-induced peripheral neuropathy was still observed in some patients, but this can easily be avoided by modulation of the administered dosage [47,48].

Many other nanocarriers have been developed, approved and marketed with the aim of improving the use of PTX. Indeed, Lipusu^®^ represents the first injectable liposomal formulation approved since 2006 by the Chinese Food and Drug Administration [32,49]. PTX was entrapped within the phospholipid bilayer made up of a mixture of lecithin and cholesterol (87:13 wt) and the vesicular systems were characterized by a mean diameter of 400 nm [50]. The addition of a 5% solution of mannitol and lysine favored the development of a freeze-dried formulation [50]. Lipusu^®^ was safer than Taxol^®^; in fact, during the in vivo anaphylaxis studies carried out in mice, the injection of Taxol^®^ at a dosage of 30 mg/kg resulted in the accumulation of inflammatory mediators in the lungs, leading to asthma-like symptoms in the Taxol^®^-treated group of mice, and this was not observed when Lipusu^®^ was used [51]. The in vitro anti-proliferative activity of the liposomal formulation along with the ameliorated safety profile endorsed the consistent use of this nanomedicine for the treatment of triple-negative breast cancer. Another liposomal formulation made up of (N-[1-(2,3-dioleoyloxy)propyl]-N,N,N-trimethylammonium) (DOTAP) and 1,2-dioleoyl-sn-glycero-3-phosphocholine (DOPC) 53:47 is under phase III clinical trial evaluation [46,52,53].

Polymeric micelles are other systems that have been proposed for the delivery of PTX. Genexol PM^®^, Nanoxel^®^ and Paclical^®^ are copolymer-based micelles developed for this purpose. The addition of 30 mg of PTX to 150 mg of monomethoxy poly(ethylene glycol) (mPEG) (2000 g/mol) and poly(DL-lactide) (PDLLA) (1750 g/mol) as diblock copolymers resulted in 20–50-nm micelles that have been marketed in Europe and South Korea since 2007 under the name of Genexol PM^®^ [53,54]. As reported by Kim and colleagues, Genexol PM^®^ and Taxol^®^ showed a similar cytotoxic activity when a concentration of 1 µg/mL of drug was used [54]. Moreover, the micellar formulation exhibited a three-fold higher maximum tolerated dose (MTD) with respect to Taxol^®^ (20 and 60 mg/kg for Taxol^®^ and Genexol PM^®^, respectively) [54]. 

Several other experimental investigations have been carried out, and currently, various phase I and II clinical trials are in progress with the aim of evaluating the eligibility of Genexol PM^®^ for the treatment of bladder, ovarian and pancreatic carcinomas (NCT01426126, NCT02739633 and NCT02739529, respectively) [53]. NK105 is an amphiphilic, core–shell type of micellar formulation made up of PEG and esterified polyaspartate, developed to favor the entrapment of PTX within the lipophilic cores of the nanostructures [55,56]. The encouraging results obtained by a phase III clinical trial performed in 2016 may lead to the contribution of this nanoformulation to the management of recurring and metastatic breast cancer (NCT01644890) [46,53]. 

Nanoxel^®^ is another type of micelle-based formulation made up of pH-sensitive N-isopropyl acrylamide (NIPAM) polymerized with vinyl pyrrolidone (VP) [57]. The system was designed to selectively release the PTX when the pH reaches acidic values, as are found in the tissues of several types of solid tumors [58,59].

Recently, the active compound was entrapped in micelles obtained through the use of XR-17 technology (Paclical^®^) developed by Oasmia Pharmaceuticals. This compound is made up of two retinoic acid derivatives, N-(all-trans-retinoyl)-L-cysteic acid methyl ester sodium salt and N-(13-cis-retinoyl)-L-cysteic acid methyl ester sodium salt. It has been observed that micelles of between 20 to 60 nm in diameter favor the intravenous administration of PTX [46,52]. A phase III clinical trial (NCT00989131) was performed by Vergote and coworkers with the aim of investigating the clinical efficacy and safety profile of Paclical^®^ [60]. The outcome of the co-administration of Paclical^®^-carboplatin was compared to what resulted when Taxol^®^-carboplatin were associated. The data demonstrated that the micellar formulation requires a shorter infusion time and pretreatment of the patient with corticosteroids is not necessary, as must be carried out when Taxol^®^ is used [60]. Interestingly, Paclical^®^ is the only formulation of PTX recommended for the treatment of animal malignancies, since its use in veterinary medicine under the trade name Paccal-Vet^®^-CA1 has been conditionally approved by the FDA. During recent experimentation, its anti-tumor activity was encouraging, when assessed in vitro on canine hemangiosarcoma cell lines [46,61,62].

## 2. Natural Polymer-Based Hydrogels in Anti-Cancer Therapy

Since their original use as contact lens starting materials, hydrogels have been increasingly employed [63]. Hydrogels are defined as tridimensional networks made up of hydrophilic polymers, capable of interacting with and retaining large amounts of water and/or biological fluids as a consequence of the presence of many functional moieties in their polymer chains [64,65]. These features allow them a certain degree of mechanical flexibility, similar to that of the extracellular matrix, and, often, a considerable degree of biocompatibility [66,67,68]. Rheological characterization of the viscoelastic properties of a gel system provides information concerning the solid- or liquid-like properties of the sample, expressed by the following two parameters: the elastic or storage modulus (G‵) and the viscous or loss modulus (G‵‵), respectively [69,70].

Based on these parameters, a polymer can behave as a viscoelastic, solid or viscous material [65]. The sol-gel transition occurs when the solid-like behavior prevails over the viscous one. By plotting the variation of G‵ and G‵‵ as a function of the frequency applied, the intersection of the two plots depicts the gelation point [71]. A gelation temperature in the range of 30–36 °C is preferable, especially for in situ gelling systems [68,72]. 

Hydrogels can be classified as a function of different parameters, i.e., crosslinking (physical/chemical), responsiveness to different stimuli (temperature, pH, light, pressure and ionic strength), porosity (nanogel and microgel), composition (homopolymer and copolymer) and origin (natural, synthetic or semi-synthetic) [68,73]. All of these parameters should be considered during the design of a hydrogel-based delivery system. Indeed, they can be modified with the aim of tailoring the physico-chemical features of the formulation, as a function of the required therapeutic outcome [74]. In this regard, nanogels are suitable for systemic injection, while microgels can be surgically implanted or orally administered [75]. 

In order to be defined a drug delivery system, a formulation should retain the entrapped compound(s), promoting a gradual/controlled/sustained release as a function of time [76]. In this regard, among different factors, it has been reported that the network size of a hydrogel plays a critical role in the choice of the administration route as well as in the modulation of the release kinetic of the entrapped compound [74,77]. 

The structure of a hydrogel is related to the association of polymer chains as well as the nature of their interaction, and the medium defines the type of hydrogel while affecting its mechanical properties. The exploitation of chemical crosslinking agents, such as glutaraldehyde (GLA) or diisocyanate, allows the formation of tridimensional networks with pronounced mechanical strength and prolonged degradation resistance as a consequence of the covalent bonds established between the polymer and the crosslinker. Considering the strength of the chemical interactions occurring in their structure, these types of hydrogels are defined as “permanent or chemical hydrogels” [78,79]. On the other hand, the addition of such chemical additives raises some toxicological concerns because some residues such as initiators or catalyzers may remain in the final formulation. The development of physical hydrogels requires mild reaction conditions and relies mainly on the inter- and intramolecular association of polymer chains that can be easily started by heating/cooling the polymer solution [80]. 

The design of thermo-sensitive hydrogels has been widely investigated as a feasible strategy for permitting sustained drug release in the required body compartment, reducing the possibility of side effects. In this regard, pluronics—and particularly pluronic F127 (F127)—are the most widely known thermo-responsive polymers, of which the sol-gel transition at body temperature has allowed the development of several formulations [68,81,82,83,84]. Unfortunately, pluronics are characterized by a weak mechanical strength and non-biodegradability; for this reason, the search for feasible alternative polymers is always in progress [85]. 

In this regard, natural polymers, such as proteins and polysaccharides, have emerged as attractive sources of versatile starting materials because of their ease of supply, low cost and non-immunogenicity. The presence of multiple functional groups promotes numerous drug–polymer interactions and the conjugation of the tridimensional matrix with appropriate ligands; moreover, their degradation occurs under physiological conditions and is characterized by the formation of safe bioproducts [86,87]. 

The gelling and bioadhesive features of proteins and polysaccharides have been used to obtain injectable in situ hydrogels for the delivery of anti-tumor compounds. Namely, their sol-gel transition temperature can be exploited in order to allow their facilitated administration as liquid “solutions” with the subsequent assembly of a tridimensional matrix at body temperature. Moreover, their adhesive features can favor a suitable residence time of the entrapped active compound(s) and proper drug concentrations, avoiding the side effects of the conventional systemic administration of other formulations, i.e., nanosuspensions and solutions.

The exploitation of injectable hydrogels in anti-cancer therapy has witnessed a consistent increase due to their soft nature and the possibility of modulating the release of the entrapped compound. This can be both as single agents or encapsulated within nanocarriers as a function of various stimuli that can be used to limit the pharmacological effect at the required target site [88]. In particular, the translation of the use of the self-healing hydrogels ranging from tissue engineering to anti-tumor therapy has demonstrated the great versatility of these systems [89]. Despite these undeniable advantages, few hydrogel-based formulations have reached the phase of pre-clinical investigation or are available on the market for clinical application. However, new technological approaches based on the development of biohybrid hydrogels (DNA-, quantum dot- or drug conjugate-based) are promising drug delivery systems to be used in anti-cancer treatment and immunotherapy [88,90,91]. 

Considering this evidence, the use of biocompatible PTX-loaded micro- and nanoparticles embedded in polysaccharide- and protein-based in-situ-forming hydrogels will be discussed in the following sections in order to highlight the recent therapeutic developments.

## 3. Polysaccharide-Based Hydrogels

### 3.1. Hyaluronic Acid-Based Hydrogels

Hyaluronic acid (HA) is a mucopolysaccharide made up of repeated monomeric units of D-glucuronic acid and N-acetyl-D-glucosamine, linked together by alternating β-1,3 and β-1,4 glycosidic bonds. As the main constituent of the extracellular matrix, it is highly biocompatible [92]. Protonation of the carboxyl moieties of D-glucuronic acid at physiological pH levels confers consistent polarity to HA and enhances its water solubility.

The biological functions of HA are related to its molecular weight. HA with a low molecular weight (LMW-HA) has immunomodulatory properties and takes part in receptor-mediated signaling, while HA having a high molecular weight (HMW-HA) has anti-inflammatory and anti-angiogenic features [93,94]. The presence of functional moieties favors the conjugation of HA to several drugs, a technological approach widely used in cancer therapy. This allows an increase in the uptake of the active compounds into the tumor mass overexpressing certain receptors of the polysaccharide, such as the cluster determinant 44 (CD44) and the receptor for hyaluronate-mediated motility (RHAMM) [95,96]. 

Several HA-based hydrogels have been developed as thermosensitive systems for the delivery of anti-cancer drugs [97,98], but to the best of the authors’ knowledge, only one formulation has been described, and this is made up of PTX encapsulated in nanocarriers embedded in HA hydrogels. In this regard, Rezazadeh and coworkers studied a thermosensitive hybrid hydrogel for the local co-delivery of PTX and doxorubicin hydrochloride (DOX). In detail, DOX was entrapped within a hydrogel made up of F127 and HA, while PTX was encapsulated within micelles of α-tocopheryl polyethylene glycol 1000 succinate (TPGS) and F127 [99]. PTX-micelles (~−10 mV and 157 nm as mean diameter) were added to the hydrogel in the form of a lyophilized powder, reaching a taxol concentration of 2, 4 or 6 mg/mL as a function of the systems initially added, while the DOX concentration was 2 mg/mL [99]. 

The inclusion of PTX-micelles in a tridimensional gel matrix gave a controlled and prolonged leakage of the lipophilic drug, while DOX was completely released in the first 12 h of analysis [99]. The addition of mixed micelles did not significantly affect the sol-gel transition time or temperature, nor the viscosity of the HA-PF127 hydrogels. The formulation was characterized by a modest degree of viscosity and good injectability as well as sustained mechanical resistance [99]. The delivery of two or more active compounds within the same formulation is a well-known anti-tumor strategy useful for obtaining pharmacological synergism and able to maximize the therapeutic efficacy of the drugs [100,101,102,103]. However, additional studies concerning in vitro/in vivo anti-tumor efficacy need to be carried out in order to validate the clinical relevance of the proposed formulation. 

### 3.2. Gellan Gum-Based Hydrogels

Gellan gum (GG) is an anionic, FDA-approved, water-soluble microbial-derived polysaccharide consisting of multiple tetrasaccharide units made up of two moieties of β-D-glucose, one of β-D-glucuronic acid and one of α-L-rhamnose [104]. Structurally, the gelation mechanism involves a temperature-driven coil-to-helix transition and, as in the case of alginates, requires the presence of cations, so it is defined as ionotropic. In this regard, it has been demonstrated that the valence of the cations strongly affects the gel strength and stability, being higher with divalent cations and lower with monovalent ones as a result of the different chemical interactions occurring [105,106]. According to the gelation mechanism proposed by Tako, intermolecular calcium bonds between the carboxyl oxygen of D-glucuropyranosyl moieties and multiple hydrogen bonds between L-rhamnopyranosyl residues occur during the sol–gel transition [107]. GG is a promising biomaterial with a great potential for application in tissue engineering, regenerative medicine, ocular drug delivery and the formulation of in situ gels [108]. Guhasarkar and colleagues developed a formulation characterized by liposomes embedded in a triggerable GG hydrogel, proposed for the intravesical delivery of PTX [109]. The choice of GG as polymer depends on its ion-triggered gelation, which can be duly exploited as a function of the target site. In detail, PTX was encapsulated within soya phosphatidylcholine liposomes and added to a gellan gum solution (0.1% *w*/*w* of GG in saline solution). The resulting formulation was characterized by appropriate injectability features and shear thinning behavior, while opposite results were observed when the concentration of the gelling polymer exceeded 0.2% *w*/*w* [109]. In vivo studies showed a 24-h intravesical retention of the formulation and a consistent cellular uptake of liposomes embedded in the GG hydrogel that were able to reach the inner bladder wall. The liposomes alone remained confined to the superficial layers of the bladder mucosa (Figure 3). These results were obtained thanks to both the mucoadhesive features of GG and its ability to form a porous matrix that enhanced the residence time of PTX-liposomes [109]. 

The anti-tumor efficacy of the formulation was assessed against rat and human urinary bladder cancer cells (NBT-II and T24, respectively). It was demonstrated that NBT-II cells were sensitive to PTX-liposomes embedded in GG hydrogels in a nanomolar range as compared to liposomal PTX (IC_50_ 55.7 ± 13.0 nM and 2.6 ± 2.03 μM, respectively), while no significant differences were observed with the T24 cells. Additional studies concerning the co-administration of drugs with different physicochemical properties will be performed in order to take full advantage of the therapeutic efficacy of the proposed formulation [109]. 

### 3.3. Alginate-Based Hydrogels

Alginate is an anionic polysaccharide extracted from brown algae cell walls that is widely used in the biomedical and alimentary fields [110]. This biopolymer is made up of β-D-mannuronic acid monomers (M) linked to ɑ-L-guluronic acid moieties (G) through 1-4 bonds, and they are arranged under different configurations, namely repeated G monomers, repeated M monomers or alternating M and G units [111]. Molecular weight, degree of acetylation, pH and M/G ratio represent the main factors affecting the physico-chemical properties and the cytocompatibility of the resulting alginate-based gels [112]. It was demonstrated that larger amounts of M-monomers can promote the production of cytokines, although this is not totally due to mannuronic monomers but to the presence of impurities such as heavy metals [113,114]. In order to bypass this potential drawback, purification processes are required before using the formulation [114]. 

Alginate hydrogels can be obtained by means of various mechanisms including thermal gelation, free radical polymerization, chemical crosslinking and cryogelation. Among these, the addition of cations (ionotropic gelation) and pH variations are the most common approaches [112,115]. Ionotropic gelation occurs when the sodium ions of the G-moieties are replaced by divalent or trivalent ions that act as crosslinking agents promoting a rearrangement of the structure known as the “egg-box structure”. This is when G-moieties of opposite sites interact with each other and arrange themselves, forming an inner hydrophilic cavity in which the oxygen interacts with the calcium ions available, rearranging into an egg-like assembly [116]. When a pH value below the pKa of both M and G monomers is achieved, pH-assisted gelation occurs. That is, the protonation of the carboxylic moieties promotes both the formation of new hydrogen bonds and an increase in the viscosity of the solution [116]. 

Yoshihoka and coworkers developed a sodium alginate (SA)-based hydrogel loaded with spray-dried hydroxyapatite microparticles (HAp) for the controlled release of PTX [117]. SA hydrogels were obtained through the ion crosslinking method, by means of the addition of CaCl_2_ 2M and HAp-PTX that were successively added to the alginate solution in a concentration range between 10 and 40% *w*/*t*. The amount of entrapped PTX was between 2.4 and 7.3% *w*/*t* and it did not influence the mean sizes of the microparticles (1–20 µm) [117]. The SA hydrogels maintained their structural integrity for up to 90% of strain applied, while the tridimensional matrix prevented the burst effect of PTX that is observed with plain Hap [117].

For the local treatment of glioma, Ding and coworkers proposed a nanocomposite made up of nanoparticles containing PTX and a DNA-alkylating agent, temozolomide (TMZ), embedded in a thermo-responsive hydrogel [118]. The rationale of using the association of the two active compounds was based on the evidence of the promotion of autophagy-related phenomena in cancer cells [119]. The double-emulsion method was used to obtain PTZ–TMZ nanoparticles made up of polyethylene glycol-dipalmitoyl phosphatidylethanolamine and calcium phosphate. A non-dosage-dependent synergism between PTX and TMZ was observed in vitro against glioma C6 cells when the two drugs were co-loaded into the particles at a ratio of PTX:TMZ 1:100 *w*/*w* [120]. For this reason, this nanoformulation was embedded within a hydrogel matrix made up of a blend of hydroxypropyl methyl cellulose (HPMC), PF127 and SA [120]. This strategy overcame some characteristic drawbacks of the various polymers when they were used as single materials. Indeed, the weak mechanical properties of PF127 together with its fast dissolution rate in biological fluids—as well as the hydrophilicity of SA and HPMC—may be critical because they could compromise both the bioadhesion and the prolonged release of the entrapped compounds [65,120]. 

The anti-proliferative synergism of the two drugs on the C6 cells was demonstrated by their enhanced autophagic death which was confirmed by the increased production of the autophagy marker LC3-II and autophagosomes in C6 tumor-bearing mice [118]. The combination of surgery and the local administration of the gel formulation into the tumor cavity promoted a high local drug concentration in the target site, avoiding the proliferation of the surviving cancer cells as well as the side effects of traditional chemotherapy [118].

### 3.4. Chitosan-Based Hydrogels

Chitosan is a cationic polysaccharide derived from the deacetylation of chitin in alkaline conditions. It is made up of N-acetyl-d-glucosamine and D-glucosamine moieties linked by β-1→4 bonds [121,122] (Figure 4). The degree of acetylation regulates the amount of free amino groups as well as the degree of water solubility, which is enhanced in acid environments due to the protonation of these moieties. The hydroxyl groups act as crucial residues for crosslinking or co-polymerization processes [123]. As reported for other biomaterials, the biodegradability of a polymer is strongly influenced by its physico-chemical features such as hydrophilic/lipophilic balance, crystallinity and molecular weight [124]. Chitosan can be degraded via chemical (acid-catalyzed degradation) or enzymatic reactions (lysozyme- or chitinase-sustained degradation) and it has been reported that these enzymes can hydrolyze the biopolymer by either randomly cleaving an inner glycosyde bond (endo-action) or by attacking a terminal residue (eso-action) [125]. The degree of substitution is a critical factor for the degradation rate, as reported for different chitosan derivatives such as hydroxyethylacryl-chitosan and N-acetylated chitosan [126,127].

The bio- and muco-adhesive features of chitosan have been used for applications such as topical drug delivery [128,129] treatment of inflamed mucosa [130,131,132] and intracellular delivery of genetic material due to the favorable affinity of DNA for cationic polymers [133,134,135].

The sol-gel transition of chitosan solutions can be obtained both through physical and chemical processes. A physical process occurs when experimental parameters such as pH or temperature are varied, or through the addition of counterions, while a chemical process takes place through the addition of chemical crosslinkers such as glutaraldehyde or diisocyanate [70]. The addition of such chemical agents modifies the chemical properties of pure chitosan-based hydrogels but can give rise to some toxicological concerns; in this regard, the use of biocompatible agents such as genipin may be a suitable approach [136,137].

Ruel-Gariépy and colleagues developed an injectable thermo-responsive in-situ-forming PTX-loaded hydrogel, proposed for the management of post-surgical resection sites. This was done with the aim of inhibiting plausible local tumor re-growth due to remaining cancer cells and, thus, avoiding systemic chemotherapy [138]. The starting material was BST-Gel^®^, the Byosintech proprietary platform made up of chitosan and β-glycerophosphate, which undergoes a sol-gel transition once body temperature is reached. This technology and its features were exploited for the controlled delivery of PTX [138]. As previously discussed, this is a promising strategy able to avoid the drawbacks of systemic chemotherapy, ensuring a considerably high local drug concentration [118]. Moreover, it is interesting to consider the proposed anti-tumor activity of chitosan as a result of the direct and indirect mechanisms involved, such as the induction of apoptosis, the inhibition of the glycolytic pathway and the modulation of immune and inflammatory mediators such as macrophages, leukocytes, interleukin 1 (IL-1) and IL-2 [138]. This anti-proliferative activity is a peculiar feature of chitin, chitosan and other polysaccharides such as β-glucans [139,140,141].

Wang and coworkers chemically modified chitosan with acrylamide and N-isopropylacrylamide with the aim of developing a thermo-responsive hydrogel to be used as adjuvant treatment with conventional chemotherapy for the treatment of colon carcinoma [142]. The heat-triggered release of PTX from the hydrogel demonstrated the potential efficacy of this formulation to exploit hyperthermia and increase the selective tumor localization of the active compound [142]. However, the delivery of PTX by means of a hydrogel raised certain issues, mainly due to the hydrophobic nature of the drug and its crystallization and precipitation in aqueous media, which lead to a decrease in its cellular uptake and anti-tumor activity [143,144].

Mahajan and coworkers proposed the nanoencapsulation of PTX (6 mg/mL) into soy phosphatidylcholine liposomes embedded in chitosan-based hydrogels as a useful approach for by-passing this problem [145]. The addition of increasing amounts of dibasic sodium phosphate (DSP) to the chitosan solution favored a decreased gelation up to a concentration of 9% *w*/*v*. Greater amounts of DSP furnished no additional benefits, so 2% *w*/*v* of chitosan and 9% *w*/*v* DSP were employed for the development of the hydrogels. The gelation of the empty hydrogels occurred at body temperature after only a few minutes and the entrapment of the PTX-liposomes inside them did not affect this parameter. A sustained and prolonged drug release was obtained. In fact, ~30% of the PTX-loaded entrapped liposomes had been released during the first 24 h, while the plain liposomes showed an almost complete leakage of PTX (88.7%) after the same period [145]. The different trend is obviously due to the presence of the polymeric tridimensional matrix, characterized by a spongy structure, the small pores of which are able to modulate the release rate of the nanocarriers and, consequently, of their payload.

An in vivo comparison of the pharmacokinetic parameters between the chitosan-based hydrogel formulation and the commercial Taxol^®^ demonstrated an improved residence time of PTX when it is entrapped in liposomes embedded within the polymeric matrix. In detail, an increased half-life (~15 h, while only 4 h for the liposomes/chitosan gel and Taxol^®^, respectively), a better drug concentration at the tumor site and a reduced infiltration of mast cells were observed, confirming the enhanced efficacy of the described hydrogels with respect to the commercial Taxol^®^ [145].

As previously reported, chitosan-based hydrogels are characterized by scarce mechanical strength. In order to overcome this drawback, Jiang and coworkers developed an in situ injectable hybrid formulation containing polyvinyl alcohol (PVA) and GLA as physical and chemical crosslinkers, respectively [146]. The PTX was encapsulated within the obtained tridimensional matrix following its inclusion in hydroxypropyl-β-cyclodextrins (HP-β-CD) (PTX: HP-β-CD molar ratio 1:2). The addition of PTX complexed with cyclodextrins promoted an increased gelation temperature of the samples (~31 and ~36 °C for empty and PTX-CD hydrogels, respectively). This result is related to the greater energy required to destroy the hydrogen bonds between the PVA and the biopolymer, as well as those between the chitosan and PTX-CD [146].

The inclusion of PVA and GLA in the same system improved the mechanical properties of the chitosan hydrogels with respect to the samples prepared with the compounds added as single agents. This result was due to the reversible hydrogen bonds between the hydroxyl- and amino groups of PVA and chitosan, respectively, and to the permanent association of aldehydic residues of GLA and the amine moieties of the biopolymer (Schiff base) [143]. The use of cyclodextrins circumvented the appearance of precipitated PTX in the tridimensional polymeric structure, and ~80% of the active compound had been released only after 13 days (Figure 5) [146]. 

The anti-tumor activity of the PTX-CD hydrogel formulation was evaluated in vivo on H22-bearing mice and compared with Taxol^®^. Namely, 10 days after the intratumoral administration of 15 mg of drug/kg of each formulation, the tumor growth was three and nine times greater, respectively, as compared to the initial tumor volume of 100 mm^3^. Moreover, the survival rate of the animals was prolonged up to 33 days in the case of the hydrogels with respect to ~20 days of the Taxol^®^-treated group. The results highlight the potential application of the hybrid chitosan-PVA-GLA crosslinked hydrogel as an in situ, injectable intratumoral system for the treatment of solid tumors [146].

Yoo and coworkers developed a chemical, crosslinked hydrogel from a novel derivative made up of glycidyl methacrylate (GM) and glycol chitosan (GC) (GM-GC 1% *w*/*v*). This mixture was converted into a gel by means of a rapid blue light irradiation (10 seconds) [147]. The formulation has shown great versatility as a matrix for a wound healing compound and a stimulating factor for bone formation [148,149]. Recently it was used to entrap PTX-β-CD complexes, evaluating the rheological features as well as the in vitro and in vivo anti-tumor activity against ovarian cancer of the resulting system [150].

The light-induced gelation of GC was confirmed by the variation in the storage modulus as a function of the frequency applied (0–100 Hz). In particular, an increase in the elastic character of the hydrogels containing PTX-β-CD was confirmed by the G’ values that moved from 1–5 up to 24 Pa after only 10 seconds of light irradiation [150]. The GC hydrogels were not degraded after immersion in a PBS solution, and a biphasic release kinetic of PTX characterized by an initial burst effect resulted [150]. In vivo experiments carried out on a SKOV-3-bearing mouse model showed that the intratumoral administration of hydrogels containing PTX promoted an increased anti-proliferative activity with respect to free PTX and PTX-β-CD. Moreover, hematoxylin and eosin staining showed a wide necrotic tumor area after 7 days when GC hydrogels had been injected, while only a partial effect was observed in the case of the use of the free drug and PTX-β-CDs (Figure 6) [150].

Recently, Pesoa and colleagues embedded poly-lactide-co-glycolic-acid (PLGA) microparticles containing PTX in pure chitosan hydrogels, proposing the formulation as a potential injectable in situ depot system for the treatment of mammary adenocarcinoma [151]. The microparticles were obtained through the emulsion method and added to a chitosan solution as a lyophilized powder. The optimized formulation, used for in vivo experiments on M-234 murine tumor models, was characterized by 1.3 mg of drug/g of the hydrogel. SEM images demonstrated the integration of the PLGA microparticles within the chitosan matrix and a prolonged drug release for up to 7 days with no burst-effect phenomena observed [151]. In vivo experiments showed that after 1 week of treatment, a single intratumoral injection of the PTX hydrogel decreased the tumor mass by ~60%, while the commercial formulation of the drug (10 mg/kg) required four administrations to promote a 40% reduction in the volume [151]. It was also interesting to observe a certain anti-proliferative activity of the empty chitosan-based hydrogels, confirming the results previously reported by Ruél-Gariépi and coworkers [138].

## 4. Protein-Based Hydrogels

Protein gelation involves denaturation and aggregation processes. In detail, the first phenomenon is based on the perturbation of the native polymer conformation (unfolding), resulting in the exhibition of new reactive sites that are successively exploited in the aggregation step to promote various interactions, novel structures of greater molecular weight and then an ordered network [86,152,153,154]. As was previously reported in the case of polysaccharides, the variation of several experimental approaches can be used to trigger the unfolding of the protein, such as chemical forces (ionic strength, enzymes) or physical parameters (temperature, pressure) [155,156].

### 4.1. Bovine Serum Albumin-(BSA) Based Hydrogels

Serum albumins have emerged as versatile drug delivery systems thanks to their notable water solubility, non-immunogenicity and ability to bind both exogenous and endogenous molecules including hormones, fatty acids, hydrophilic/lipophilic compounds, genetic material and peptides [157,158]. Bovine serum albumin (BSA), also known as “fraction V” of cow serum, is an animal-derived protein with a molecular weight of ~66 kDa, an isoelectric point of 4.7 and a sequence homology of ~76% with the human isoform [159,160]. The protein is characterized by natural fluorescent properties as a consequence of two tryptophan residues (Trp 134 and 213) and multiple tyrosine (Tyr) and phenylalanine (Phe) moieties (Figure 7) [161]. This peculiar feature can be exploited to investigate BSA–drug interactions, as was true for the anti-tumoral compounds neratinib [162] and linifanib [163] as well as for the antimycotic iprodione [164]. 

The gelation mechanism of BSA involves a rearrangement of the tertiary structure of the protein due to the unfolding of certain polypeptides. This is followed by the occurrence of several protein–protein interactions that, in turn, lead to the formation of a protein network, usually described as a cage-like structure in which solvent or host molecules can be entrapped [160,165]. 

Qian and coworkers developed an injectable hydrogel formulation made up of PEG-modified BSA for the delivery of red blood cell membrane-based nanoparticles containing PTX (RBC-PTX NPs) [166]. In detail, BSA was esterified with low-molecular-weight PEG (200 g/mol), and the time required for the sol–gel transition was evaluated as a function of different BSA-PEG concentrations. Since a gelation time of ~7 min was observed when 16% *w*/*v* of modified BSA was used, and lower amounts of polymer did not promote the formation of the hydrogel, this concentration was chosen for the various experimental investigations [166]. The addition of RBC-PTX NPs favored an increase in the gelation time (up to ~10 min) and a prolonged release of the drug was obtained (36% after 1 week in PBS 0.01 M, pH 7.4, 37 °C) [166]. The in vitro degradation of the hydrogel in simulating body fluids showed that ~50% of the initial weight of the formulation was maintained after 50 days, while the in vitro anti-tumor efficacy of the formulation was assessed in a human gastric carcinoma cell line (MKN-45), demonstrating a decrease of ~40% in the cell viability when a concentration of 20 ng/mL of PTX was used [166]. These encouraging results were confirmed by in vivo evaluation of the formulation tested on MKN-45 tumor-bearing mice. In fact, a tumor growth inhibition of ~65% and an enhanced necrosis of the tumor tissue were obtained as a consequence of the better drug localization in the neoplastic area. Moreover, the healthy organs such as the liver, spleen, lung and heart revealed no structural changes, and a reduced number of liver metastatic lesions was also obtained, confirming the efficacy and safety of the proposed formulation as an innovative system for the administration of PTX [166].

### 4.2. Collagen-Based Hydrogels 

Collagen is a ubiquitous fibrous biomaterial fundamental for the maintenance of the integrity of cells and tissues [167]. To date, 29 types of collagen have been identified, with type I collagen being the most abundant as the leading component of corneas, heart, bone and extracellular skin matrices [168]. Each collagen molecule consists of three intertwined polypeptide chains arranged under a right-handed helix responsible for the tensile and mechanical strength of the biopolymer [169]. This helical conformation requires the presence of a repetitive motif in its core, namely Gly-X-Y-, in which glycine (Gly) residues are repeated at every third position while the X and Y positions are made up of proline and hydroxyproline moieties. The presence of these is critical for the stability of the collagen structure, due to the formation of intermolecular hydrogen bonds [170,171].

Watanabe and coworkers developed PTX-loaded hyaluronic acid nanoparticles (HA-NPs) embedded in a collagen-based hydrogel (CH) and evaluated the cytotoxicity of the formulation against highly and poorly metastatic breast cancer cell lines (MDA-MB-231 and MCF-7, respectively) [172]. These nanocarriers were obtained through the ethylene-diamine-tetra-acetic acid (EDTA)-assisted hydrothermal procedure, and PTX was adsorbed onto the particle surfaces only when an ethanol:water ratio of 5.5:4.5 was used, as a consequence of its high lipophilicity. It was interesting to observe that MDA-MB-231 cells treated with PTX-HA-NPs embedded in CH showed a greater decrease in their viability (~60%) with respect to the MCF-7 cells [172]. Indeed, the IC_50_ value was ~6 µM for MDA-MB-231 cells while it was undetectable for the MCF-7 cells. This trend was explained as a consequence of the intrinsic ability of the highly metastatic MDA-MB-231 cells to secrete matrix metalloproteinases able to metabolize the tridimensional collagen matrix and promote the release of the drug molecules [172].

Recently, Wang and coworkers proposed a collagen-based hydrogel obtained using polyvinyl alcohol (PVA) as crosslinker that was used to retain PLGA nanoparticles containing PTX. This was done with the aim of developing an implantable scaffold to be used for the treatment of residual cancer cells after the surgical resection of tumor masses, as well as a reconstitution implant to replace excised mammary tissue (Figure 8) [173]. 

The collagen matrix was characterized by a highly porous matrix with a compression modulus of 33 KPa (at 40% strain), able to support the necessary tissue growth and to promote a prolonged release of PTX. In vitro tests demonstrated that only ~20% of the 50 mg of PTX contained in the formulation was released within the first 48 h, while a prolonged drug leakage was observed for up to 260 h. Moreover, high concentrations of the active compound were found in the adipose tissue of mice treated with the hydrogel formulation as compared to the plasma compartment, which avoided the potential toxicity exerted by the drug on other healthy organs [173].

The efficacy of the formulation was also corroborated by its intraperitoneal administration in MDA-MB-231 tumor-bearing mice, showing a significant reduction in the tumor volume and a complete regression of the tumor masses in several animals. Additional studies on these hybrid, self-crosslinking hydrogels containing PTX as potential implants for breast cancer resections will be performed in order to evaluate their real application in clinical practice [173].

#### Gelatin-Based Hydrogels

Gelatin is a hydrophilic polymer obtained from the hydrolysis of collagen. Based on acid or alkaline experimental conditions, two types of polymers can be obtained, namely cationic A-type gelatin or anionic B-type gelatin [174]. Indeed, the different isoelectric points of these two (6.5–9 and 4.8–5.2, respectively) should be considered during the development of gelatin-based formulations because they can influence the retention of active compounds [174,175]. The presence of the peculiar arginine-glycine-aspartate tripeptide motif (RGD) favors cell adhesion and proliferation, while the presence of MMP degradation sites allows its degradation [176]. Moreover, the noteworthy biocompatibility and the generally recognized as safe (GRAS) status of the polymer promote its exploitation as a biomaterial for tissue engineering and drug delivery applications. It is also used for the treatment of bleeding in the form of a hemostatic compressed sponge called Gelfoam^®^ [177]. Gelatin is characterized by a gel-sol transition at body temperature, and its gelation mechanism involves the reconstitution of a triple collagen helix [153].

Zhang and coworkers proposed the co-encapsulation of PTX and tetrandrine (Tet) within mPEG-polycaprolactone (mPEG-PCL) nanoparticles embedded in a crosslinked, gelatin-based hydrogel, with the aim of exploiting the pharmacological effects of the active compounds in the peritumoral tissues of gastric cancer [178]. The use of crosslinkers or chemical functionalization are common procedures used to modulate the scarce mechanical features of gelatin [78]. The idea was to exploit the potential synergism between PTX and the natural pro-oxidant (Tet) with the aim of promoting the oxygen reactive species (ROS)-mediated apoptotic pathway in cancer cells, whereas a gelatin-based matrix was used to allow the prolonged release of both compounds into the tumor site [178]. Namely, the injection of the hydrogel should be followed by the gradual release of mPEG-PCL nanoparticles as well as the entrapped drugs from the biopolymer matrix, due to its melting temperature (35 °C).

In vitro characterization of the system showed that a 50% leakage of the active compounds had taken place after 5 days, and significant cell death was observed when the formulation was incubated with two gastric cancer cell lines (BGC-823 and SGC-7901). In particular, it was interesting to note that the addition of N-acetyl cysteine (used as an antioxidant and ROS scavenger compound) scarcely affected the cytotoxic effect of either of the drugs, demonstrating that their efficacy was mainly due to an enhanced production of ROS [178]. These results are remarkable, because an enhanced antioxidant capacity of certain types of cancer cells is a well-known mechanism of resistance against PTX [179,180,181,182].

The in vivo administration (via laparotomy) of gelatin-based hydrogels containing PTX- and Tet-loaded nanoparticles (7.5 and 15 mg/kg, respectively) demonstrated a considerable decrease in both the volume and weight of a gastric tumor with respect to the free form of the drugs and empty mPEG-polycaprolactone nanoparticles. This result was mainly due to the direct administration of the gelatin hydrogel onto the tumor; indeed, once the biopolymer-based structure was subject to disorganization, the nanocarriers were able to promote the introduction of a high concentration of PTX and Tet into the peritumoral tissue. Moreover, Western blotting analysis showed an increased expression of pro-apoptotic Bax protein and a decreased expression of Bcl-2. A significant inhibition of BGC-823 cell migration was also observed during a wound scratch assay. The results demonstrated that Tet can efficiently sensitize gastric cancer cells to PTX and suggests that their association (oxidation therapy) may represent a conceivable approach for the management of gastric cancer. However, additional studies are required before the clinical translation of this formulation can come about [178,179]. 

Recently, Vigata and coworkers developed an implantable delivery system made up of gelatin functionalized with methacrylic anhydride (GelMA) for the controlled release of Abraxane^®^ [183]. The aim here was to obtain a suitable formulation to be used following surgical mastectomy or lumpectomy in order to prevent breast cancer recurrence [183]. The rheological properties of GelMA hydrogels prepared using 5, 10 and 15% *w*/*v* of the polymer were not affected by the inclusion of Abraxane^®^ when 1.07 and 2.65 µg of nanoformulation per µl of hydrogel were used. On the contrary, the GelMA concentration influenced both the encapsulation efficiency as well as the in vitro release profiles of Abraxane^®^ [183]. Namely, 10% *w*/*v* of GelMA was shown to be a good compromise between entrapment yield and release sustainability. Indeed, GelMA hydrogels prepared using the aforementioned concentrations of Abraxane^®^ were used to evaluate their cytotoxicity on MDA-MB-231 cells. That is, the smallest amount of Abraxane^®^ (i.e., 1.07 µg/µL) promoted a decrease in cell viability similar to that obtained when greater concentrations of Abraxane^®^ were used. This phenomenon was justified as the result of the formation of particle aggregates characterized by mean sizes greater than the pores of the hydrogels, which prevented an efficacious interaction between the nanosystems and the cells [183].

### 4.3. Silk Fibroin (SF)-Based Hydrogels 

Silk fibroin (SF) is a fibrous biopolymer produced by several insects such as spiders and pseudo-scorpions, but *Bombyx mori* silkworms’ SF is actually the most employed derivative [184]. SF extracted from silkworm cocoons is characterized by an inner core of fibroin molecules (75% *w*/*w*) made up of a heavy and a light chain (350 and ~26 kDa, respectively) linked together by disulfide bonds and an external layer of hydrophilic glue-like moieties of sericin. Although sericin can elicit an immune response, its hydrophilicity facilitates its removal following a degumming process (boiling treatment in alkaline solutions) [185,186]. The heavy chain of SF comprises a large amount of hydrophobic repetitive domains (Gly−Ala−Gly−Ala−Gly−Ser)_n_ alternated by hydrophilic random short aminoacidic sequences arranged under a series of stacked crystalline β-sheets, responsible for the mechanical properties and high tensile strength of SF [187].

The light chain contains several hydrophilic aminoacidic residues such as glycine, serine and aspartate and contributes to the elasticity of SF [186]. As reported by Matsumoto and coworkers, the sol-gel transition of SF is usually a slow process that starts with minimal conformational changes and proceeds with the formation of irreversible β-sheet assemblies. The pH, temperature and polymer concentration marginally affect the structure of SF-based gels, which suggests that a major role is played by the rearrangement of the hydrophilic and hydrophobic portions of the protein [188].

Wu and coworkers described the development of SF nanoparticles containing PTX and the antibiotic salinomycin (Sal) as single agents, embedded into ultrasound-assisted SF hydrogels. The goal of the study was to exploit the co-delivery of the two drugs in order to enhance the anti-proliferative activity of PTX, with the aim of killing stem and non-stem cancer cells [189]. A previous characterization of nanoparticles showed that ~50% of the initial 5 mg of PTX added during the preparation procedure was retained by the colloidal structures when an SF:PTX ratio of 25:5 *w*/*w* was used [190]. In this recent study, 6 mg of Sal and 15 mg/mL of biopolymer were used to obtain Sal-loaded SF nanoparticles (entrapment efficiency of the drug equal to 35%) [189].

Empty and nanoparticle-enriched SF hydrogels were characterized by similar injectability and structural integrity in PBS. The in vitro release studies showed that the release of the active compounds was influenced by the physico-chemical features of each drug; namely, PTX was gradually released for up to 30 days, while the leakage of Sal was complete after 24 h [189]. Local administration of the SF hydrogel containing PTX and Sal-loaded SF-nanoparticles (18 and 4 mg/kg of PTX and Sal, respectively) in subcutaneous murine hepatic cancer (H22) models showed significant results in terms of the reduction in the relative tumor volume, the animal survival rate and the inhibition of CD44^+^CD133^+^ stem cell spreading. Better efficacy with co-delivery of the two active compounds with respect to the formulations containing the drugs as single agents was also demonstrated. Additional investigations will be carried out in order to assess the clinical efficacy of the proposed injectable hydrogel for loco-regional chemotherapy [189].

## 5. Conclusions

Biocompatibility, biodegradability and non-toxicity are mandatory requirements in the design and development of an innovative drug delivery system [191,192]. In this regard, natural polymers are feasible alternatives to synthetic and semi-synthetic raw materials [193,194]. To date, Oncogel^®^, a thermosensitive PLGA-PEG-PLGA copolymer-based hydrogel, is the only formulation ever proposed for the local release of PTX in solid tumors, but it brought about no significant improvements in terms of anti-tumor efficacy, so it did not reach the market [195,196]. Exploiting the gelling and bio/mucoadhesive features of various polysaccharides and proteins for obtaining formulations to be used for the local administration of a bioactive compound is a reasonable strategy to use in order to maximize the therapeutic efficacy of different molecules, especially when the site-specific application of an anti-tumor drug is required. In this review, an overview concerning the exploitation of polysaccharide- and protein-based injectable in situ hydrogels containing biocompatible PTX-loaded micro- and nanocarriers was provided (Table 1 and Table 2). Despite the encouraging results, much investigation concerning the in vivo behavior and pharmacological efficacy of these formulations will be required in order to promote a rapid translation from bench to bedside and manage novel therapeutics for the treatment of tumors.

## Figures and Tables

**Figure 1 gels-07-00033-f001:**
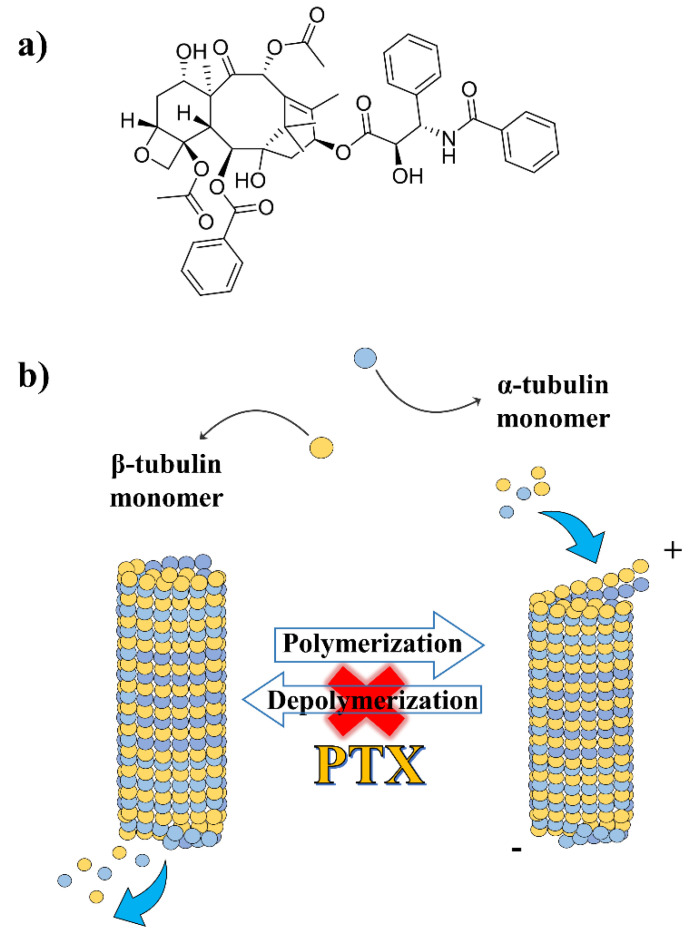
Chemical structure of paclitaxel (PTX) (**a**) and schematic representation of its mechanism of action (**b**).

**Figure 2 gels-07-00033-f002:**
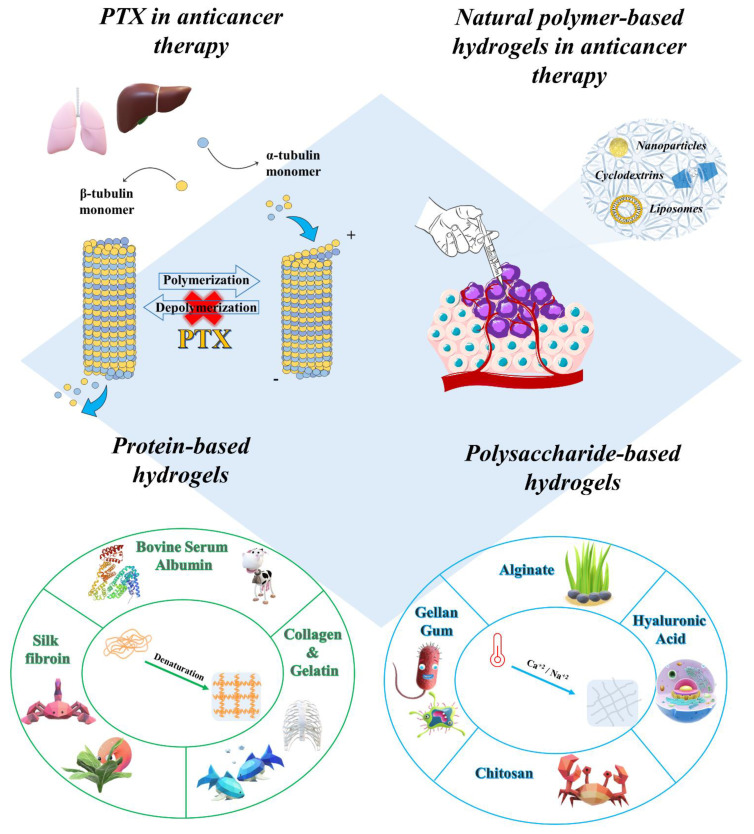
Graphical representation of the area covered in the manuscript.

**Figure 3 gels-07-00033-f003:**
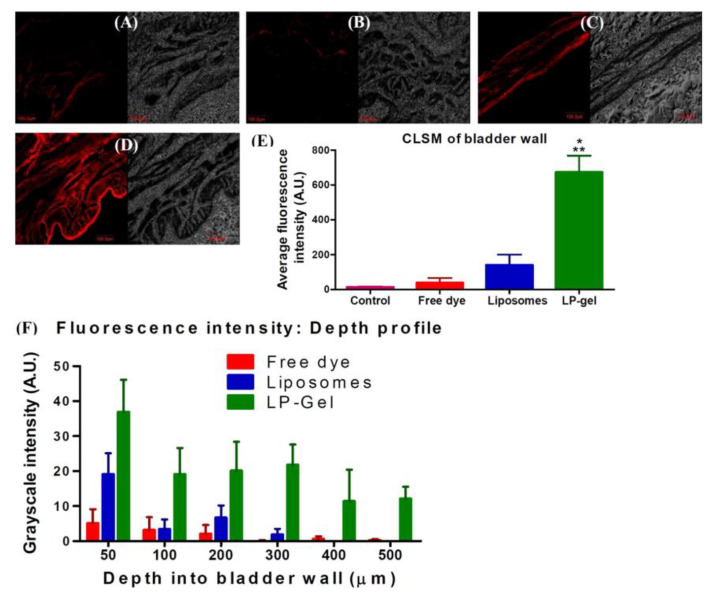
Confocal laser scanning micrographs (CLSMs) of cross-section of bladder wall (red filter and bright field): (**A**) Control bladder; (**B**) bladder with free dye (rhodamine-6G); (**C**) bladder with dye-loaded liposomes and (**D**) bladder with dye-loaded LP-Gel (scale bar for all images = 100 μm). Analysis of CLSM images: (**E**) average fluorescence intensities (* *p* < 0.05 compared to free dye, ** *p* < 0.05 compared to liposomes) and (**F**) depth profile of fluorescence intensities (*p* < 0.05 from two-way ANOVA). Reproduced with permission from [109], Elsevier (2017).

**Figure 4 gels-07-00033-f004:**
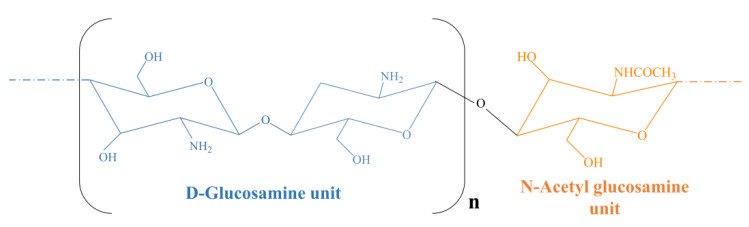
Chemical structure of chitosan.

**Figure 5 gels-07-00033-f005:**
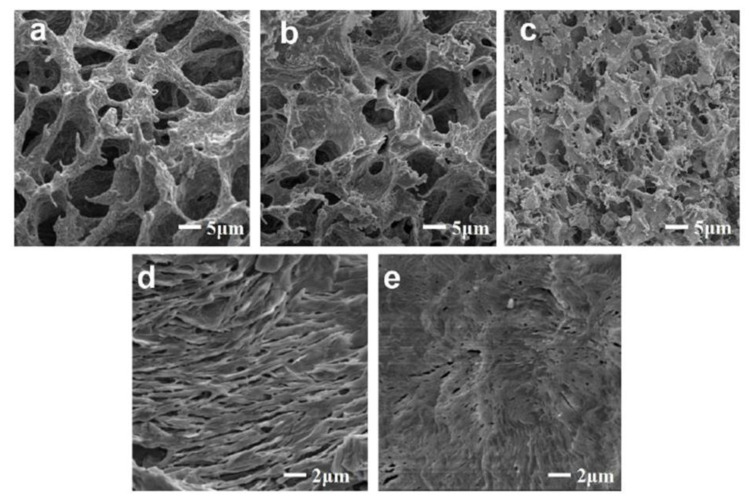
Scanning electron microscopy (SEM) images of (**a**) pure chitosan (CS)-based hydrogels; (**b**) CS-based hydrogels crosslinked with polyvinyl alcohol (PVA) hydrogels; (**c**) CS-based hydrogels crosslinked with glutaraldehyde (GLA); (**d**) CS-based hydrogels crosslinked with GLA and PVA; (**e**) PTX–cyclodextrin (PTX-CD) complexes embedded in CS-based hydrogels crosslinked with GLA and PVA. Reproduced with permission from [146], Elsevier (2016).

**Figure 6 gels-07-00033-f006:**
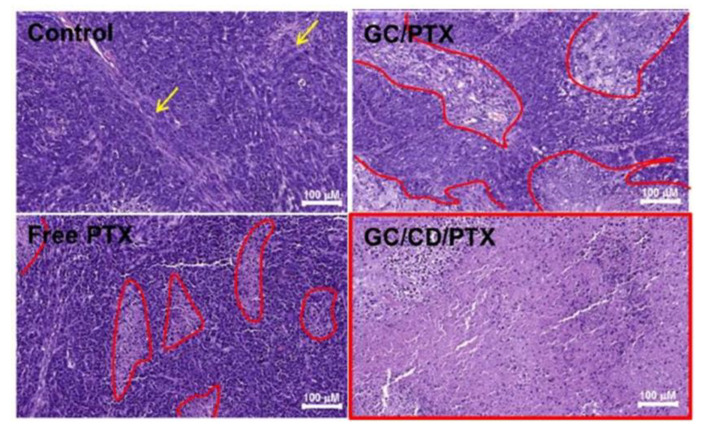
Intratumoral administration of PTX, glycol chitosan (GC)/PTX and GC/CD/PTX in a Scheme 3. bearing mouse model: hematoxylin and eosin-stained images of tumor masses after 7 days. In the control sample, the yellow arrows indicate necrotic tissues induced by the limited expansion of the tumor tissue in the small animal. In the free PTX-, GC/PTX- and GC/CD/PTX-treated samples, the red lines indicate necrotic tissues induced by apoptosis. Adapted with permission from [150], MDPI (2019).

**Figure 7 gels-07-00033-f007:**
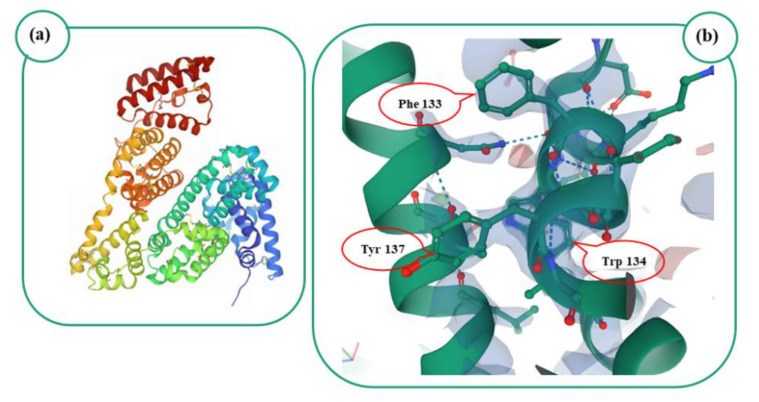
(**a**): Image from the RCSB Protein Data Bank (PDB) of PDB ID 4F5S [161], IUCr (2012). (**b**): Image of tryptophan (Trp), phenylalanine (Phe) and tyrosine (Tyr) residues occurring in 4F5S, Mol*.

**Figure 8 gels-07-00033-f008:**
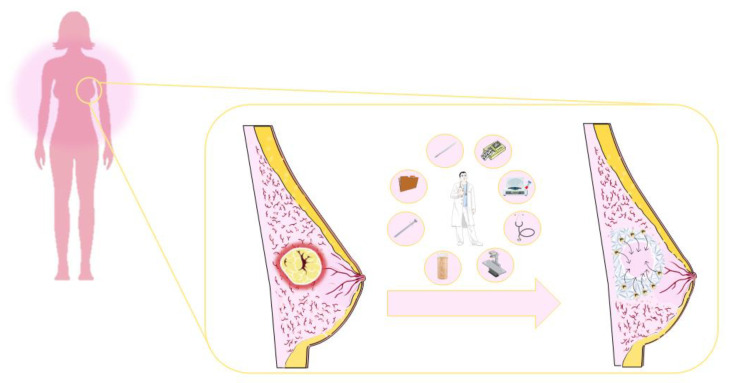
Schematic representation of the clinical application of PTX-loaded poly-lactide-co-glycolic-acid (PLGA) nanoparticles embedded in collagen-based hydrogels.

**Table 1 gels-07-00033-t001:** Composition and physico-chemical properties of PTX-loaded micro/nanosystems embedded in polysaccharide-based hydrogels.

Polymer	Carrier Composition	Physico-chemical Features	Hydrogel Composition	Proposed Application	Main Results	Reference
HA	Micelles P127/TPGS 7:3 *w*/*w* mixture; 3 mg of PTX	MS: ~160 ± 20 nm PdI: ~0.30 ± 0.10 ZP: ~−10 ± 1 mV	PF127 20% *w*/*v* HA 2% *w*/*v* DOX 2 mg/mL	Treatment of solid tumors	Sustained release of PTX from mixed micelles embedded in HA-PF127 hydrogels.	[99]
GG	Liposomes PTX:lipid 1:2 *w*/*w*	MS: 124 ± 7 nm PdI: 0.22 ± 0.001 ZP: −17 ± 2 mV	0.1% *w*/*v* of gellan gum in saline solution	Intravesical delivery	Cation-triggered gelation.	[109]
SA	HAp Microparticles PTX 2.4-7.3 *w*/*v*	MS: 4 ± 0.2 µm PdI: / ZP: /	SA 0.5-3% *w*/*t* SA:HAp-PTX 10-40% *w*/*t*	Treatment of bone cancer	HAp microparticles embedded in alginate matrix can act as both drug carrier and filler for the cancerous, resected bone tissue.	[117]
HPMC-SA-PF127	Nanoparticles mPEG-DPPE-CaP-NP PTX:TMZ 1:100 *w*/*w*	MS:38 ± 1 nm PdI: 0.16 ± 0.02 ZP: −40 ± 1 mV	/	Treatment of glioma	Thermo-responsive hydrogel.	[118]
CS	Liposomes SP 4.25% *w*/*v* PTX 0.6% *w*/*v* SD 0.75% *w*/*v* S80 0.75% *w*/*v*	SD-Liposomes MS: 176 ± 6 nm PdI: 0.3 ZP: −16 ± 1 mV	CS 2% *w*/*v* DSP 4-12% *w*/*v*	Localized drug delivery system	CS hydrogel containing PTX-liposomes showed improved localization of the drug in the target site and improved safety as compared to Taxol^®^.	[145]
		S80-Liposomes MS 136 ± 4 nm PdI: 0.3 ZP: −19 ± 0.5 mV				
CS-PVA-GA	Inclusion complex HP- β-CD 0.1% *w*/*v* PTX: HP-β-CD 1:2 *w*/*w*	/	CS 2% *w*/*v* CS:PVA 1:1 *w/w* GA 220 µM	Intratumoral treatment of solid cancers	Combination of chemical and physical crosslinking resulted in modified CS hydrogels with considerable mechanical strength. The modified porous matrix allowed the sustained release of PTX into the tumor site for 13 days.	[146]
CS-Glycerol	β-CD complexes PTX 1 mg 2 mM β-CD 3 mg 2 mM	/	GC 1.5 g GM 7 mg	Treatment of ovarian cancer	Improved PTX solubility after its inclusion in β-CD host cavity.	[150]
CS	PLGA Microparticles PTX 1.30 mg PTX/g formulation PVA 2% *w*/*v*	MS: ~6 ± 0.13 µm PdI: / ZP: /	/	Treatment of mammary adenocarcinoma	Improved anti-tumor efficacy of chitosan-based hydrogel with respect to PTX solution.	[151]

β-CD: β-cyclodextrins; CS: chitosan; GG: gellan gum; GM: glycidyl methacrylate; HA: hyaluronic acid; SA: sodium alginate; mPEG-DPPE-CaP: polyethyleneglycol-dipalmitoylphosphatidyl-ethanolamine; CaP-NP: calcium-phosphate nanoparticles; HAp: hydroxyapatite; P127: Pluronic F127; PTX: paclitaxel; TPGS: D-alpha-tocopheryl polyethylene glycol-1000 succinate; TMZ: temozolomide; SP: soy phosphatidylcoline; DSP: dibasic sodium phosphate; MS: mean sizes; PdI: Polydispersity Index; ZP: zeta potential.

**Table 2 gels-07-00033-t002:** Composition and physico-chemical features of PTX-loaded micro/nanosystems embedded in protein-based hydrogels.

Polymer	Carrier Composition	Physico-chemical Features		Hydrogel Composition	Proposed Application	Main Results	Reference
PEG-BSA	RBC nanoparticles	MS: ~130 ± 2 nm PdI:/ ZP:~ −10 ± 1mV		PEG-BSA 16% *w*/*v*	Treatment of gastric cancer	Sustained release of PTX at the tumor site with decrease in intraperitoneal metastasis.	[166]
Collagen	HAp nanoparticles	/		PTX-HAp NPs 2% *w*/*v* Collagen: high concentration medium:alkali solution 8:1:1 *v*/*v*	Treatment of metastatic cancer	Improved anti-tumor efficacy against MDA-MB-231 cells.	[162]
Collagen-PVA	PLGA nanoparticles PLGA 500 mg PTX 50 mg PVA 0.625% *w*/*v*	MS: ~62 nm PdI: 0.22 ± 0.022 ZP:~−30 mV		PVA 5% Collagen 10 mg/mL PLGA-PTX NPs 5% *w*/*v*	Breast cancer	Injection of collagen-PVA hydrogel for PTX release in the tumor-resection cavity and as tissue implant to replace the cancerous mammary tissue.	[173]
Gelatin	PEG-b-PCL 20 mg	MS: 70–90 nm PdI:/ ZP: /		Gelatin 500 mg PTX and Tet 0.5 mg/mL	Treatment of gastric cancer	Tet improved the anti-proliferative activity of PTX.	[178]
GelMA	Abraxane^®^ 1.07 µg/µL and 2.65 µg/µL per µl of hydrogel formulation	/		GelMA 5-10-15% *w*/*v*	Prevention of breast cancer recurrence after mastectomy and/or lumpectomy	GelMA improved the cytotoxic activity of Abraxane.	[183]
SF	SF-based nanoparticles Sal 6 mg PTX 5 mg	Sal-SF	MS: ~240 nm PdI: 0.15 ZP: ~−15 mV	SF 15 mg/mL SF/PTX 25:5 *w*/*w*	Locoregional chemotherapy	Increase in Sal maximum tolerated dose; co-administration of SF-based nanoparticles encapsulating PTX and Sal as single agents in SF-based hydrogels improved their anti-tumor efficacy against CD44^+^CD133^+^ cancer stem cells.	[189]
		PTX-SF	MS: 158 ± 0.4 nm PdI: 0.12 ± 0.02 ZP: ~−3 mV				

BSA: bovine serum albumin; HAp: hydroxyapatite; RBC: red blood cell nanoparticles; PLGA: poly-lactic-co-glycolic acid; PVA: polyvinyl alcohol; Tet: tetrandrine; GelMA: gelatin methacryloyl; MAA: methacrylic anhydride; SF: silk fibroin; Sal: salinomycin; MS: mean sizes; PdI: Polydispersity Index; ZP: zeta potential.

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
