# Peer review of "Recent Advances of Taxol-Loaded Biocompatible Nanocarriers Embedded in Natural Polymer-Based Hydrogels"

_gels, 2021, doi:10.3390/gels7020033_

Round 1
Reviewer 1 Report
- It is a very interesting review. However, the last sentences of the abstract are a bit confusing. Kindly review them.
- I think it would be better if the word 'natural-based hydrogels' was renamed as 'natural-polymer-based hydrogel'.
- The paragraph on the cytotoxic mechanisms of PTX seems unnecessary. The whole point of the paper is based on the physicochemical properties of the drug and has nothing to do with the mechanism of action of the drug. The authors might want to reconsider keeping that information.
Author Response
Please, see the attachment

Reviewer 2 Report
The review article by Voci et.al., discussing the recent advances of hydrogel-embedded biocompatible nanocarriers in cancer treatment is an interesting piece of academic/research document that is timely as it will help the researchers to get an in-depth understanding about the recent developments in the field. Although the manuscript is well written and covers most of the important domains, this reviewer feels that the authors omitted a few important references (a few examples are given below). The review may be published once the authors update the article with all relevant references and discussions within the article.
- Designing hydrogel for controlled drug delivery by J. Li and D.J. Mooney (Nat Rev Mater. 2016,1(12),16071) is a significant publication in the field and must be cited and discussed in the manuscript, preferably at the beginning of section 2.
- Another article that must be discussed in the same section is a review published by R. Narayanaswamy and V.P. Torchilin in 2019 (Molecules 2019, 24, 603; doi:10.3390/molecules24030603).
- A recent article by S. Kumar and A. Bajaj (Biomater. Sci., 2020,8, 2055-2073) also deserves discussion and citation in this review article.
Author Response
Please, see the attachment

Reviewer 3 Report
This is a very nice manuscript including a hot topic that would be very useful for cancer investigations. It is very well written and has losts of specific references.
Suggestions:
-An figure/scheme including sections in the manuscript should be provided
-In line 47, it should be employed the term pharmacophore to refer to necessary moieties to maintenance its anticancer activity.
-In line 31 the structure pf PTX should be included
-The 1994 Corey synthesis of taxol should be included in references
-A figure explaining the role of PXT mechanism of action related to cytotoxic would be useful
-a chemical structure of chitosan should be included in section 3.4- in section 3.4 chould be mentioned the degradation profile of chitosan and derivatives.
-In line 635, Zhang and coworkers proposed the co-encapsulation of PTX and tetrandrine (Tet) 635 within mPEG-polycaprolactone (mPEG-PCL) nanoparticles embedded in a crosslinked, 636 gelatin-based hydrogel. Please explain in more detail the advantages of this system.
Author Response
Please, see the attachment

Reviewer 4 Report
I would like to thank you for the opportunity to review the manuscript: Recent Advances of Taxol-Loaded Biocompatible Nanocarriers Embedded in Natural Polymer-Based Hydrogels I consider this manuscript very well written, clear and consistent.
The paragraphs was well planned and executed. The data are consistent and important for the development of a novel pharmaceutical formulation with excellent activity against cancer cells. The scientific results were well discussed and the bibliographical references are relevant and current. Finally, I believe the manuscript is of interest to the readers of the journal.